

# A SNP variation in an expansin (*EgExp4*) gene affects height in oil palm

Suthasinee Somyong[1], Phakamas Phetchawang[1], Abdulloh Kafa Bihi[1,2], Chutima Sonthirod[1], Wasitthee Kongkachana[1], Duangjai Sangsrakru[1], Nukoon Jomchai[1], Wirulda Pootakham[1] and Sithichoke Tangphatsornruang[1]

[1] National Omics Center, National Science and Technology Development Agency (NSTDA), Klong Luang, Pathum Thani, Thailand
[2] School of Life Sciences and Technology, Institut Teknologi Bandung, Bandung, Indonesia

## ABSTRACT

Oil palm (*Elaeis guineensis* Jacq.), an Aracaceae family plant, is utilized for both consumable and non-consumable products, including cooking oil, cosmetics and biodiesel production. Oil palm is a perennial tree with 25 years of optimal harvesting time and a height of up to 18 m. However, harvesting of oil palm fruit bunches with heights of more than 2–3 meters is challenging for oil palm farmers. Thus, understanding the genetic control of height would be beneficial for using gene-based markers to speed up oil palm breeding programs to select semi-dwarf oil palm varieties. This study aims to identify Insertion/Deletions (InDels) and single nucleotide polymorphisms (SNPs) of five height-related genes, including *EgDELLA1*, *EgGRF1*, *EgGA20ox1*, *EgAPG1* and *EgExp4*, in short and tall oil palm groups by PacBio SMRT sequencing technology. Then, the SNP variation's association with height was validated in the Golden Tenera (GT) population. All targeted genes were successfully amplified by two rounds of PCR amplification with expected sizes that ranged from 2,516 to 3,015 base pair (bp), covering 5′ UTR, gene sequences and 3′ UTR from 20 short and 20 tall oil palm trees. As a result, 1,166, 909, 1,494, 387 and 5,384 full-length genomic DNA sequences were revealed by PacBio SMRT sequencing technology, from *EgDELLA1*, *EgGRF1*, *EgGA20ox1*, *EgAPG1* and *EgExp4* genes, respectively. Twelve variations, including eight InDels and four SNPs, were identified from *EgDELLA1*, *EgGRF1*, *EgGA20ox1* and *EgExp4*. No variation was found for *EgAPG1*. After SNP through-put genotyping of 4 targeted SNP markers was done by PACE™ SNP genotyping, the association with height was determined in the GT population. Only the mEgExp4_SNP118 marker, designed from *EgExp4* gene, was found to associate with height in 2 of 4 height-recordings, with $p$ values of 0.0383 for height (HT)-1 and 0.0263 for HT-4. In conclusion, this marker is a potential gene-based marker that may be used in oil palm breeding programs for selecting semi-dwarf oil palm varieties in the near future.

Corresponding author
Suthasinee Somyong,
suthasinee.som@nstda.or.th

## INTRODUCTION

More than 2,500 palm species in the Arecaceae (Palmmae) family are widely grown in tropical and sub-tropical regions of the world as ornamental and economic plants. Oil palm is one of the economic palm species, which include coconut palm (*Cocos nucifera*), date palm (*Phoenix dactylifera*) and African oil palm (*Elaeis guineensis*). The *Elaeis* genus contains only two species, *E. oleifera (American oil palm)* and African oil palm (*Henson, 2012*). African oil palm or oil palm is classified as three types, including Dura, Tenera and Pisifera, according to their shell thickness. Tenera oil palm is widely grown as commercial oil palm because of its high yield. Oil palm is a perennial species that is widely grown in Africa and Asia. It can be harvested for up to 25 years and some varieties can reach heights of up to 18 m in good conditions (*Barcelos et al., 2015*). Along with high fruit bunch production, semi-dwarf height is another favorable trait to be included in the oil palm variety improvement. There have been several interspecific crosses between *E. guineensis* and *E. oleifera*, such as the COMPACT lines (*Alvarado & Henry, 2015*), to make palm hybrids with dwarf loci of *E. oleifera*. Alternatively, some palm breeders prefer to do intraspecific crossing (*Rajanaidu et al., 2000*) rather than using *E. oleifera* because of its lower yield production compared to *E. guineensis*. To reduce harvesting labor and facilitate mechanization, improvement of oil palm in both yield traits and vegetative traits, such as reduced trunk height and bunch stalk length has been integrated (*Barcelos et al., 2015*; *Yaakub et al., 2020*). Oil palm crosses with Dumpy dura can produce low truck height, like crosses between Deli dura with Dumpy AVROS pisifera (*Soh et al., 1981*; *Nair, 2010*). An oil palm population with Nigerian origin showed short stem height (*Rajanaidu & Jalani, 1994*; *Breure, 2006*). Oil palm progenies with La Mé pisifera origin also showed short stem height and longer bunch stalk (*Ithnin & Kushairi, 2020*). Marker assisted selection (MAS) can speed up the conventional oil palm breeding process in oil palm. Therefore, gene-based markers designed from height-related genes in oil palm (*E. guineensis*) can be used to speed up oil palm variety improvement.

This research aims to develop gene-based molecular markers targeting height-related genes, including *EgDELLA1*, *EgGRF1*, E*gGA20ox1*, *EgAPG1* and *EgExp4*. These gene variations were first elucidated by PacBio sequencing and only the SNP markers were genotyped and analyzed for height association. These genes were targeted because of their reported relation to height (*Boonkaew et al., 2018*; *Choi, Kim & Kende, 2004*; *Lee et al., 2015*; *Pootakham et al., 2015*; *van der Knaap, Kim & Kende, 2000*). In oil palm, height QTLs on chromosome 10, 14 and 15, control 10–21% of phenotype variance expected (PVE) for height. DELLA and GA2 oxidase, on chromosome 14, were suggested as potential height genes (*Pootakham et al., 2015*). Moreover, a major QTL on chromosome 16 was reported to control 51% of PVE for height and asparagine synthase was proposed as a potential height gene (*Lee et al., 2015*). Recently, our team confirmed that *EgDELLA1* (*Somyong et al., 2019*) and *EgGRF1* (*Somyong et al., 2020*) are height-associated genes in the GT population. As mentioned above, we targeted *EgDELLA1* (DELLA), *EgAPG1* (asparagine synthase) and *EgGRF1* (growth regulating factor). The other genes,

*EgGA20ox1* (gibberellin 20 oxidase) and *EgExp4* (expansin), were targeted as height-related genes in other species, including coconut (*Boonkaew et al., 2018*) and rice (*Choi, Kim & Kende, 2004*).

## MATERIALS AND METHODS

### Plant material and phenotype details

The same GT oil palm population that was previously reported (*Somyong et al., 2020*) was used in this study. The population samples were kindly provided by the Golden Tenera Company Limited, Krabi, Thailand. The GT population contained 180 individuals that were planted in 2008. The population resulted from 30 crosses between six female parents and five male parents. The female parents (dura fruit type) consisted of A 43/9D, A 1/2D, R 15/14D, R 8/9D, R 10/1D and R 10/5D while the male parents (pisifera fruit type) consisted of R 9/8P, R 5/21P, R 3/8P, KA 17/2P and R 16/7P. Height (HT) was recorded four times in three successive years from 8 to 10 year oil palm: in March 2016, October 2016, March 2017 and November 2018.

### DNA extraction

DNA samples were extracted from oil palm leaves by using a DNA extraction kit, DNeasy Plant Mini Kit (QIAGEN, Germantown, MD, USA) and using a CTAB/Chloroform-Isoamyl alcohol DNA extraction technique (*Cullings, 1992*). A mixture of 1 liter CTAB solution contained 1M Tris pH 8.0, 5M NaCl, 0.5M EDTA, and 20 g of CTAB (Cetyl Trimethyl Ammonium Bromide). PVP and β-mercaptoethanol were freshly added to the CTAB solution before it was used for DNA extraction. Quality and concentration of DNA were evaluated by agarose gel electrophoresis and a NanoDrop™ 1000 Spectrophotometer (Thermo Fisher Scientific, Fitchburg, WI, USA).

### Oil palm samples, PCR amplification and barcoding preparation for PacBio SMRT sequencing

Forty oil palm individuals of the GT population, including 20 short oil palm individuals and 20 tall oil palm individuals, were selected for PacBio SMRT sequencing, based on their height phenotypes at the 4 recorded times. To obtain full-length genomic DNA sequences of five targeted genes, including *EgDELLA1*, *EgGRF1*, *EgGA20ox1*, *EgAPG1* and *EgExp4*, specific primers were designed at 5′ UTR and 3′ UTR sites, using Primer3 (http://bioinfo.ut.ee/primer3-0.4.0/) (Table S1). The PCR amplification was performed by Phusion High-Fidelity DNA Polymerase (Thermo Fisher Scientific, Baltics UAB, Lithuania) and ran on a Veriti® 96-Well Thermo cycler (Applied Biosystems, Waltham, MA, USA). Two round PCR amplification reported in previous work (*Pootakham et al., 2017*), with modification, was performed to obtain the barcoded amplicons, which were used for SMRT PacBio sequencing. For the 1$^{st}$ round of PCR amplification by specific primers tagged with M13F and M13R (Table S2), 10–20 ng of genomic DNA was used in a 20 μl PCR volume. The 20 μl PCR volume consisted of 1 unit (U) of Phusion High-Fidelity (HF) DNA Polymerase, 1× Phusion HF Buffer, 0.2 mM dNTPs, 1.5 mM MgCl$_2$, 0.1 μM of each primer, and distilled water. The PCR conditions for amplification were the

following; initial denaturation at 98 °C for 30 s and then 35 cycles of denaturation at 98 °C for 10 s, annealing at 55 °C for 30 s, elongation at 72 °C for 2 min, and an ending step at 72 °C for 10 min. The PCR products from the 1st round of PCR were used as the DNA template for the 2nd round of PCR. For the 2nd round of PCR by M13F and M13R primers tagged with 16-base PacBio barcodes (Table S3), 2 µl of the 1st round PCR product was used as the template in 50 µl PCR volume. A final volume of 50 µl consisted of 1 U of Phusion HF DNA polymerase, 1× Phusion HF Buffer, 0.2 mM dNTPs, 1.5 mM MgCl$_2$, 0.1 µM of each primer, and the remaining volume of distilled water. The same PCR conditions explained above were used. The DNA size of 5–10 µl of the PCR products was determined by agarose gel electrophoresis. The remaining 40–45 µl of PCR products was purified with Agentcourt AMPure magnetic beads (Beckman Coulter, Indianapolis, IN, USA), and finally diluted in 10 µl of 1× TE. The concentration of barcoded amplicons was measured using a Qubit 2.0 fluorometer and a Qubit dsDNA BR assay kit (Thermo Fisher Scientific, Waltham, MA, USA). The purified DNA was pooled in equimolar concentration in at least 500 ng of the pooled DNA (a volume of 30–40 µl). The pooled DNA was used to construct the SMRTbell libraries. The barcoded amplicons were ligated into SMRTbell adapters. All libraries were sequenced on a PacBio RSII system (Pacific Biosciences, Menlo Park, CA, USA), using the P6-C4 chemistry with 360-min movie lengths.

## Sequence data analysis

The PacBio raw reads of targeted genes, including *EgDELLA1*, *EgGRF1*, *EgGA20ox1*, *EgAPG1* and *EgExp4*, processed by PacBio RSII, were analyzed by SMRT analysis software package (version 2.3). The Circular Consensus Sequencing (CCS) reads were obtained using RS_ReadsOfInsert protocol with –minPasses 5 –minPredictedAccuracy 0.99 –maxLength 3,500 into a FASTQ file. These files were first separated by using barcodes, according to each oil palm sample, and then were grouped for each gene. This file separation or classification was performed by using Python script and the MUSCLE program for data alignment. Next, the CCS reads of each oil palm sample for each gene were used to select the full-length genomic DNA sequences. The CCS reads of the full-length genomic DNA sequences of each gene were pooled according to short or tall oil palm group. The grouped full-length genomic DNA sequences were mapped to the reference gene sequences using minimap2 version 2.17-r943-dirty. SNP and InDel polymorphic sites among short and tall oil palm groups were identified by GATK version 4.1.1.0. The SNPs and InDels with depth coverage of more than 20× were used for developing molecular markers for oil palm breeding programs. Details of CCS reads and the full-length genomic DNA sequences of each gene can be found at the NCBI database under accession number: PRJNA760254.

## SNP Marker designing for high through-put PACE™ SNP genotyping

This study focused only on SNP marker development. SNP primers were designed by 3CR Bioscience Co. Ltd. SNP primers including specific nucleotides at each SNP variation at the targeted genes and nucleotides required for fluorescent emission (Table S4). PCR Allelic Competitive Extension (PACE) is an allele-specific technology for SNP genotyping

that includes 2 allele-specific forward primers and 1 common reverse primer. Universal PACE™ Genotyping Master Mix (Standard ROX) (Essex CM20 2BU; 3CR Bioscience, UK) contained solution required for PCR amplification and fluorescent detection, including FAM (blue emission) and HEX (red emission) specified for each SNP change. Details of reaction preparation followed the PACE™ User Guide v1.6 (www.3crbio.com) with modification. The reaction mixes were placed in a 384-well plate that included 1–10 ng DNA template per reaction well and prepared by the wet DNA method. Five μl reaction volume comprised of 2.5 μl of DNA template, 2.5 μl of universal PACE™ Genotyping Master Mix and 0.07 μl of PACE assay mix. Universal PACE™ Genotyping Master Mix contained Tag DNA polymerase, universal fluorescent reporting cassette, dNTPs, performance enhancers, $MgCl_2$, and 5-carboxyl-x-rhodamine, succinimidyl ester (ROX). The assay mix contained allele-specific primer1-FAM, allele-specific allele2-HEX and common reverse primer in the ratio of final concentration as 0.168 μM: 0.168 μM: 0.42 μM, respectively, for each reaction. The PCR amplification was run on QuantStudio 6 Flex real-time PCR system (Thermo Fisher Scientific, Waltham, MA, USA). The PCR reaction for PACE genotyping was the following. For the 1st step, enzyme activation was conducted at 94 °C for 15 min. For the 2nd step, 10 cycles of touch down PCR were conducted by denaturation at 94 °C for 20 s, and touch down annealing and extension starting at 61 °C to 55 °C for 60 s by decreasing 0.6 °C per cycle. For the 3rd step, 25 cycles were conducted by denaturation at 94 °C for 20 s, annealing and extension at 57 °C for 60 s and an ending step at 37 °C for 60 s. QuantStudio™ Real-Time PCR Software v1.3 was used to analyze SNP genotypes, which were two separated homozygous genotypes emitting either FAM (blue) or HEX (red) and heterozygous genotype emitting both fluorescences (green).

## Population structure, statistical and association analyses

The contribution of targeted SNP markers to the phenotype traits was analyzed by ANOVA and association analysis. Descriptive statistics of height phenotype data (HT) were analyzed by SPSS 11.5. This statistic package was also used to analyze preliminary relationships between the polymorphic loci of targeted genes with HT by comparing mean height using One-Way ANOVA. The significant association of the polymorphic loci with the traits was then analyzed by TASSEL 2.1 (http://www.maizegenetics.net/#!tassel/c17q9) by GLM with Q model with 10,000 permutations that included Q-matrix. The required information for the association analysis included genotype, height phenotype and Q-matrix information. Inferred ancestry of individuals of optimal K value from STRUCTURE output was used as Q-matrix information. STRUCTURE 2.3.4 (http://pritchardlab.stanford.edu/structure.html) and STRUCTURE harvester (http://taylor0.biology.ucla.edu/structureHarvester/) were used to analyze population structure and determine optimal K value, respectively. The inferred ancestry of individuals was used as Q-matrix information by setting its value as covariance in the association analysis of the targeted markers with height. Details of STRUCTURE analysis of the GT population were already explained in our previous work (Somyong et al., 2019). Briefly, STRUCTURE was run three times at this setting: Length of Burnin Period = 50,000, number of MCMC

Reps after Burnin = 50,000 and K setting = 1–10. The optimal K was determined by using STRUCTURE Harvester (*Earl & vonHoldt, 2012*) and the optimal K calculated using the Evanno method (*Evanno, Regnaut & Goudet, 2005*). The optimal K of the GT population (180 individuals) was 3 because the highest Delta K was found for this value (*Somyong et al., 2019*). For marker information, previous markers (*Somyong et al., 2019*; *Somyong et al., 2020*), including mEgDELLA1-1, mEgDELLA1-11, mEgACCO-pr2, mEgSSRffb10-8, mEgGRF1-3 and mEgGA20ox1-4, and an additional marker, mEgExp4_SNP118 from this study, were used in TASSEL analysis. The *p* value applied to determine significance of marker-trait association was less than 0.05.

## RESULTS

### Details for height phenotype of the GT population

Height information of HT-1, HT-2, HT-3 and HT-4 for the GT population (180 individuals) was explained in our previous work (*Somyong et al., 2020*). HT-1, HT-2 and HT-3 were recorded in 6-month intervals in 8–9 year oil palm plants while HT-4 was recorded in 10-year oil palm plants. The mean height of the GT population was 192 cm for HT-1, 231 cm for HT-2, 263 cm for HT-3, and 380 cm for HT-4, with a 75 cm average increase within a year. To identify nucleotide variations of height-related genes by PacBio SMRT sequencing, 40 individuals were selected based on height phenotype. Height phenotype was classified as short and tall groups, including 20 individuals of the shortest oil palm plants and 20 individuals of the tallest oil palm plants from the GT population. Height phenotype of the same short and tall oil palm individuals was recorded for all 4 height recordings. The height distribution of the short and tall oil palm groups is illustrated in Fig. S1. For HT-1, average height was 132 cm for the short group and 254 cm for the tall group. For HT-2, average height was 164 cm for the short group and 300 cm for the tall group. For HT-3, average height was 190 cm for the short group and 342 cm for the tall group. For HT-4, average height was 289 cm for the short group and 471 cm for the tall group. The height difference between short and tall oil palm groups was 122 cm for HT-1, 136 cm for HT-2, 152 cm for HT-3 and 182 cm for HT-4. Height between short and tall groups was significantly different for the 4 height recordings with a *p* value of 0.000 by using *t*-test.

### Primer testing for amplification of full-length genomic DNA sequences of the height-related genes

The first step of this study was primer design and full-length amplification testing in short and tall oil palm groups from the GT population. Five candidate genes, *EgDELLA1*, *EgGRF1*, *EgGA20ox1*, *EgAPG1* and *EgExp4*, were targeted in this study. The targeted sites of primer design were 5′ UTR and 3′ UTR sites of the genes. The full-length genomic DNA sequences of these genes were obtained from oil palm draft sequences of the Malaysian Palm Oil Board (MPOB) (http://genomsawit.mpob.gov.my/index.php?track=30). For full-length gene amplification, two primer pairs were designed for each gene from 5′ UTR and 3′ UTR sites (Table S1). The selected primers that amplified the full-length gene

products of expected sizes, from 2,516 to 3,015 bp, included EgDELLA1-P1, EgGRF1-P2, EgGA20ox1-P2, EgAPG1-P1 and EgExp4-P1 (Fig. S2).

## Sample preparation for PacBio SMRT sequencing of the height-related genes

The 40 short and tall oil palm samples were prepared for PacBio SMRT sequencing of the *EgDELLA1*, *EgGRF1*, *EgGA20ox1*, *EgAPG1* and *EgExp4* genes. The full-lengths of these genes were successfully amplified by the M13-tagged EgDELLA1-P1, EgGRF1-P2, EgGA20ox1-P2, EgAPG1-P1 and EgExp4-P1 primers (Table S2) in the 1$^{st}$ round of PCR with bands close to the expected sizes of 3,015 bp, 2,516 bp, 2,759 bp, 2,917 bp and 2,586 bp, respectively. The PCR products from the 1$^{st}$ round of PCR were used as a DNA template for the 2$^{nd}$ round of PCR. These M13-tagged PCR products were then carried to the second amplification with several combinations of barcode-tagged M13 primer sets (Table S3). Each barcode-tagged M13 primer set consisted of forward and reverse primers that were used in the separation of each oil palm sample for each gene amplification. The results of amplification by M13-tagged primers and barcode-tagged M13 primers also had bands close to the expected sizes of 3,015 bp, 2,516 bp, 2,759 bp, 2,917 bp and 2,586 bp respectively, as shown in Fig. S3.

## Analysis of sequence variations among short and tall oil palm groups

The full-length genomic DNA sequences of *EgDELLA1*, *EgGRF1*, *EgGA20ox1*, *EgAPG1* and *EgExp4* genes of short and tall oil palm samples were revealed by PacBio SMRT sequencing technology (Table 1). For *EgDELLA1*, 37 of the 40 total oil palm plants, including 18 short and 19 tall oil palms, have been successfully sequenced, representing 1,166 full-length genomic DNA sequences with a size range of 2,959–3,246 bp.

For *EgGRF1*, 37 of the 40 total oil palm plants, including 18 short and 19 tall oil palms, have been successfully sequenced, representing 909 full-length genomic DNA sequences with a size range of 1,933–2,607 bp. For *EgGA20ox1*, 36 of the 40 total oil palm plants, including 18 short and 18 tall oil palms, have been successfully sequenced, representing 1,494 full-length genomic DNA sequences with a size range of 2,138–2,913 bp.

For *EgAPG1*, all 40 total oil palm plants have been successfully sequenced, representing 387 full-length genomic DNA sequences with a size range of 2,060–2,952 bp. For *EgExp4*, all 40 total oil palm plants have also been successfully sequenced, representing 5,384 full-length genomic DNA sequences with a size range of 2,493–2,705 bp. The results showed that the highest number of full-length genomic DNA sequences was found from *EgExp4* while the lowest number of sequences was found from *EgAPG1*. The variations of the number of full-length genomic DNA sequences results from amplification ability during the two rounds of PCR amplification.

The results of sequence variation, position and variation type of all these genes, among short and tall oil palm groups, are listed in Table 2. Variations were found for 4 of the 5 genes. No variation was found for the *EgAPG1* gene. For the *EgDELLA1* gene, there was a total of 1,166 sequences, representing short and tall oil palm groups with 578 and 677 sequences, respectively. We found 3 variations, containing 1 insertion and 2 SNPs.

**Table 1 Full-length genomic DNA sequence details for *EgDELLA1*, *EgGRF1*, *EgGA20ox1*, E*gAPG1* and *EgExp4*, using PacBio SMRT sequencing technology between the short and tall oil palm groups.**

| Gene name | Number of short oil palm trees having full-length genomic DNA sequence | Number of tall oil palm trees having full-length genomic DNA sequence | Number of barcoded sequences | Number of full-length genomic DNA sequences | Expected size of full-length reference genomic DNA sequences (bp) | Size of full-length PacBio SMRT sequences (bp) |
|---|---|---|---|---|---|---|
| *EgDELLA1* | 18 | 19 | 1,255 | 1,166 | 3,015 | 2,959–3,246 |
| *EgGRF1* | 18 | 19 | 1,092 | 909 | 2,516 | 1,933–2,607 |
| *EgGA20ox1* | 18 | 18 | 1,756 | 1,494 | 2,759 | 2,138–2,913 |
| *EgAPG1* | 20 | 20 | 1,287 | 387 | 2,917 | 2,060–2,952 |
| *EgExp4* | 20 | 20 | 5,454 | 5,384 | 2,586 | 2,493–2,705 |

**Note:**
bp = base pairs.

**Table 2 The description of variant type and position on the full-length genomic DNA sequences of *EgDELLA1*, *EgGRF1*, *EgGA20ox1*, *EgAPG1* and *EgExp4* between the short and tall oil palm groups, using PacBio SMRT sequencing technology.**

| Gene | No. of Full-length genomic DNA sequences | No. of full-length genomic DNA sequences in short group | No. of full-length genomic DNA sequences in tall group | Variation position from the forward primer position | Variation change from the reference genomic DNA sequences | Variant type | Variant change mostly found in |
|---|---|---|---|---|---|---|---|
| *EgDELLA1* | 1,166 | 578 | 677 | 312 | T/TA | Insertion | some short oil palm |
| | | | | 2,100 | T/A | SNP | some short oil palm |
| | | | | 2,248 | G/A | SNP | some short oil palm |
| *EgGRF1* | 909 | 468 | 441 | 1,044 | TATA/T | Deletion | some short oil palm |
| | | | | 1,100 | G/GT | Insertion | some short oil palm |
| | | | | 1,553 | A/ATCTC | Insertion | some short oil palm |
| | | | | 1,553 | ATCTCTC/A | Deletion | some tall oil palm |
| *EgGA20ox1* | 1,494 | 767 | 726 | 1,428 | G/GAA | Insertion | some short oil palm |
| | | | | 1,428 | G/GA,GAAA | Insertion | some tall oil palm |
| | | | | 1,468 | T/G | SNP | some short oil palm |
| *EgAPG1* | 387 | 156 | 231 | n/a | n/a | n/a | n/a |
| *EgExp4* | 5,384 | 3,249 | 2,135 | 118 | T/C | SNP | some tall oil palm |
| | | | | 989 | TC/T | deletion | some tall oil palm |

**Notes:**
n/a = no variation.
No. = number.

The position of the insertion was at 312 (T/TA), and the SNP variations were at 2,100 (T/A) and 2,248 (G/A), all of which are illustrated on the *EgDELLA1* reference gene sequence, which has a full-length of 3,015 bp (Fig. S4). For the *EgGRF1* gene, there was a total of 909 sequences, representing short and tall oil palm groups with 468 and 441 sequences, respectively. We found 4 variations, including 2 deletions and 2 insertions. The positions of deletions were at 1,044 (TATA/T) and 1,553 (ATCTCTC/A), and insertions were at 1,100 (G/GT) and 1,553 (A/ATCTC), all of which are illustrated on the *EgGRF1* reference gene sequence, which has a full-length of 2,516 bp (Fig. S5). For the *EgGA20ox1* gene, there was a total of 1,494 sequences, representing short and tall oil palm

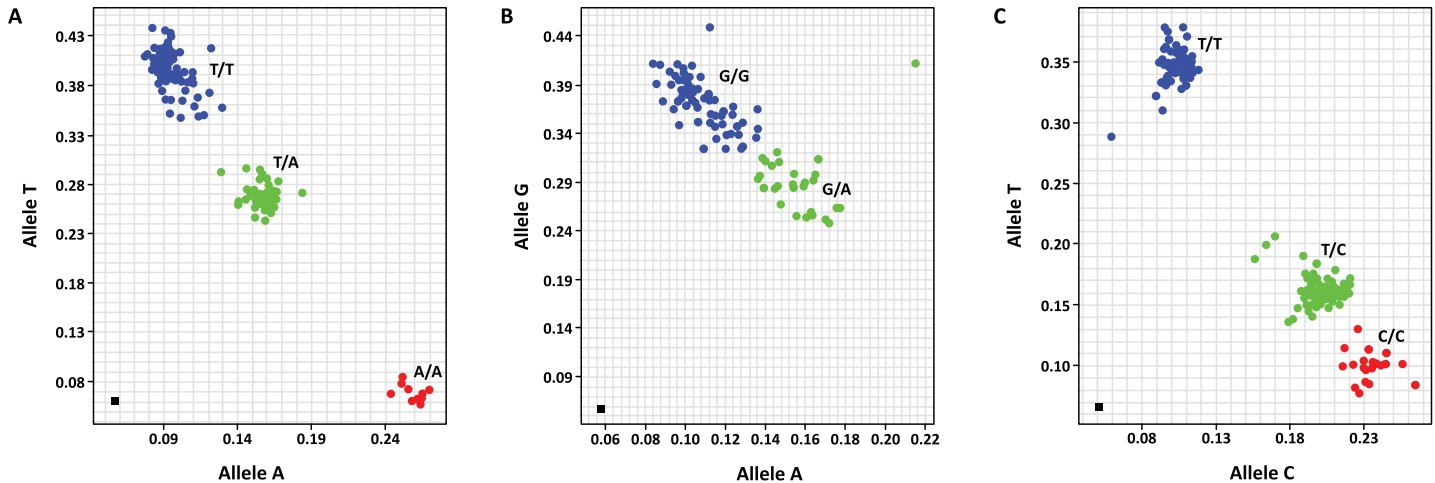

**Figure 1** Allelic discrimination plots of the polymorphic SNP markers, including mEgDELLA1_SNP2100 (genotypes T/T, T/A and A/A) (A), mEgDELLA1_SNP2248 (genotypes GG and GA) (B) and mEgExp4_SNP118 (genotypes T/T, T/C and C/C) (C) that amplified from the GT population.

groups with 767 and 726 sequences, respectively. We found 3 variations, including 2 insertions and 1 SNP. The positions of insertions were at 1,428 (G/GAA) and 1,428 (G/GA, GAAA), and the SNP variation was at 1,468 (T/G), all of which are illustrated on the *EgGA20ox1* reference gene sequence, which has a full-length of 2,759 bp (Fig. S6). For the *EgExp4* gene, there was a total of 5,384 sequences, representing short and tall oil palm groups with 3,249 and 2,135 sequences, respectively. We found 2 variations containing 1 deletion and 1 SNP. The position of the deletion was at 989 (TC/T), and the SNP variation was at 118 (T/C), both of which are illustrated on the *EgExp4* reference gene sequence, which has a full-length of 2,586 bp (Fig. S7). No variation was found for the *EgAPG1* gene, which is also illustrated on the *EgAPG1* reference gene sequence, which has a full-length of 2,917bp (Fig. S8). This may be due to either no-true variation or an inadequate amount of the full-length genomic DNA sequences that is found between short and tall groups for this gene.

## High through-put genotyping by PACE™ SNP genotyping

This work has targeted only SNP marker development. PACE™ SNP genotyping was used in this study. Four SNP primer sets were designed from three target genes, including *EgDELLA1*, *EgGA20ox1* and *EgExp4*, and are listed in Table S4. These SNP primer sets included mEgDELLA1_SNP2100, mEgDELLA1_SNP2248, mEgGA20ox1_SNP1468 and mEgExp4_SNP118. Three of the SNP primer sets, including mEgDELLA1_SNP2100, mEgDELLA1_SNP2248 and mEgExp4_SNP118, were polymorphic in the GT population while mEgGA20ox1_SNP1468 was monomorphic in the same population. The allelic discrimination plots of these polymorphic markers are illustrated in Fig. 1. Genotypes of mEgDELLA1_SNP2100 included A/A (10 individuals), T/A (65 individuals) and T/T (103 individuals) (Fig. 1A). Genotypes of mEgDELLA1_SNP2248 included G/A (51 individuals) and G/G (124 individuals) (Fig. 1B). Genotypes of mEgExp4_SNP118 included C/C (25 individuals), T/C (78 individuals) and T/T (66 individuals) (Fig. 1C).
Table 3 Mean height comparison of mEgExp4_SNP118 genotypes of 8–10 year oil palm of the GT population by using ANOVA analysis (A) and significant association of the mEgExp4_SNP118 marker by TASSEL (B).

**(A)**

| Trait | Genotype 1 | Genotype 2 | Height mean difference (cm.) between genotype 1-genotype 2 | Std. error (cm.) | Sig.* |
|---|---|---|---|---|---|
| HT-1 | C/C | T/C | 16.81(*) | 7.86 | 0.034 |
| | | T/T | 24.96(*) | 8.03 | 0.002 |
| HT-2 | C/C | T/C | 18.14(*) | 9.06 | 0.047 |
| | | T/T | 25.51(*) | 9.25 | 0.007 |
| HT-3 | C/C | T/C | 14.41 | 10.04 | 0.153 |
| | | T/T | 20.53(*) | 10.26 | 0.047 |
| HT-4 | C/C | T/C | 28.78(*) | 13.33 | 0.032 |
| | | T/T | 38.63(*) | 13.63 | 0.005 |

*The mean difference is significant (Sig.) at the 0.05 level.

**(B)**

| Trait | Locus | df_Marker | F_Marker | p_Marker | Previous work |
|---|---|---|---|---|---|
| HT-1 | mEgDELLA1-1 | 5 | 2.3511 | 0.0429* | *Somyong et al. (2019)* |
| HT-2 | mEgDELLA1-1 | 5 | 2.6198 | 0.0261* | |
| HT-3 | mEgDELLA1-1 | 5 | 2.1946 | 0.0571 | |
| HT-4 | mEgDELLA1-1 | 5 | 3.1094 | 0.0103* | *Somyong et al. (2020)* |
| HT-1 | mEgGRF1-3 | 8 | 2.2319 | 0.0276* | |
| HT-2 | mEgGRF1-3 | 8 | 2.2332 | 0.0275* | |
| HT-3 | mEgGRF1-3 | 8 | 2.0064 | 0.0487* | |
| HT-4 | mEgGRF1-3 | 8 | 2.2654 | 0.0253* | |
| HT-1 | mEgExp4_SNP118 | 2 | 3.3295 | 0.0383* | This work |
| HT-2 | mEgExp4_SNP118 | 2 | 2.5946 | 0.0778 | |
| HT-3 | mEgExp4_SNP118 | 2 | 1.5947 | 0.2061 | |
| HT-4 | mEgExp4_SNP118 | 2 | 3.7194 | 0.0263* | |

Note:
*The $p$ value of significant association was less than 0.05.

The genotypes of mEgExp4_SNP118 were found to have the most distribution of allele frequencies in the two homozygous groups and heterozygous group when compared with the other two markers.

## ANOVA and association analysis of mEgExp4_SNP118 with height

ANOVA analysis shows that SNP changes of mEgExp4_SNP118 in the GT population, from nucleotide T to C, affected height significantly while that of mEgDELLA1_SNP2100 and mEgDELLA1_SNP2248 did not. Height details and the height distribution of mEgExp4_SNP118 genotypes are shown in Table S5. We suggest that the *EgExp4* gene contributes to the height trait in the GT population. ANOVA analysis confirmed that mEgExp4_SNP118 genotypes, C/C, T/C and T/T, have significant height differences between them, for all 4 height-recordings, with $p$ values of 0.002–0.047 (Table 3A). Oil palm individuals with genotype C/C were significantly taller than individuals with

genotype T/C and genotype T/T in all 4 height-recordings, by 17–39 cm. In addition, genotypes T/C and T/T did not have significant height differences between them.

For HT-1, individuals with genotypes C/C were significantly taller than individuals with genotypes T/C and T/T by 17 cm and 25 cm, respectively. For HT-2, individuals with genotype C/C were significantly taller than individuals with genotypes T/C and T/T by 18 cm and 26 cm, respectively. For HT-3, individuals with genotype C/C were significantly taller than individuals with genotypes T/C and T/T by 15 cm and 21 cm, respectively. For HT-4, individuals with genotype C/C were significantly taller than individuals with genotypes T/C and T/T by 29 cm and 39 cm, respectively. Even though, the individuals with genotype T/C had intermediate height, they were not significantly different from the individuals with genotype T/T. We suggest that a SNP change from T to C, in position 118 of *EgExp4*, is involved in increasing height, while T acts in the opposite way by decreasing height. The contribution of *EgExp4* to height was further confirmed by association analysis using TASSEL.

To perform association analysis by TASSEL, besides the genotype and phenotype data, a Q-matrix from the GT population was needed. The Q-matrix was determined by STRUCTURE analysis and was shown in our previous study (*Somyong et al., 2019*, *2020*). The trait information consisted of HT-1, HT-2, HT-3 and HT-4. After association analysis, with the permutation value set as 10,000, we found a significant association between mEgExp4_SNP118 and height, as shown in Table 3B. The mEgExp4_SNP118 marker was found to be significantly associated with height in 2 from 4 height-recordings with *p* values of 0.0383 for HT-1 and 0.0263 for HT-4. This result supports data from the ANOVA analysis. Based on our previous work and this work, the three markers that have been shown to associate with height include mEgDELLA1-1 (*Somyong et al., 2019*), mEgGRF1 (*Somyong et al., 2020*) and mEgExp4_SNP118 that target DELLA, growth regulating factor and expansin genes, respectively.

## DISCUSSION

Semi-dwarf oil palm is beneficial in terms of reducing total harvesting time and harvesting labor costs. Understanding the genetic control of height in oil palm can be used in MAS for speeding up the breeding process in oil palm. In this work, we found that the expansin gene, *EgExp4*, along with the recently reported *EgDELLA1* (*Somyong et al., 2019*) and *EgGRF1* (*Somyong et al., 2020*) genes, are associated with height. Height is a quantitative trait that involves several genes in GA biosynthesis, such as GA 20 oxidase (*Monna et al., 2002*; *Spielmeyer, Ellis & Chandler, 2002*), the GA signal pathway, such as GA responsive repressors, DELLA genes (*Gale & Youssefian, 1985*; *McGinnis et al., 2003*; *Pearce et al., 2011*; *Reitz & Salmon, 1968*) and non GA-related pathways, such as expansin (*Choi et al., 2003*; *Xing et al., 2009*). Due to the number of genes involved in height control, several genetic markers would be needed in MAS to be successful. Thus, more height controlling, and associated, genes still need to be examined in order to increase efficiency of MAS in oil palm breeding.

In this work, 12 variations, including 8 InDels and 4 SNPs, were identified from *EgDELLA1, EgGRF1, EgGA20ox1 and EgExp4*. Three SNPs and 2 InDels were positioned

in the 5′ UTR, while one SNP and 6 InDels were positioned in the gene sequences. This suggests that an amplicon sequencing at 5′ UTR is just as necessary as the gene sequence. After confirmation in the GT population (180 oil palm individuals) using 4 SNP markers, including mEgDELLA1_SNP2100, mEgDELLA1_SNP2248, mEgGA20ox1_SNP1468 and mEgExp4_SNP118, three SNPs were found to be polymorphic (mEgDELLA1_SNP2100, mEgDELLA1_SNP2248 and mEgExp4_SNP118) and one SNP is monomorphic (mEgGA20ox1_SNP1468). One SNP marker, mEgExp4_SNP118, was found to significantly associate with height in 8–10 year oil palm plants, while the other two polymorphic SNPs were not associated with height in the same population. This suggests that the 20 short and 20 tall oil palm plants used to represent the population were not enough to determine the true amount of polymorphism by PacBio SMRT Sequencing. Because the original parental Dura and Pisifera of the GT population were Dumby dura, Deli dura and AVROS pisifera, this mEgExp4_SNP118 marker may potentially be used in MAS in other populations with the same original parental lines. Moreover, sequencing from other oil palm populations with different original parental lines, such as African dura, Yanggambi pisifera, La Me pisifera, EKONA pisifera and Calabar pisifera, may lead to the discovery of variations that may associate with height in other oil palm populations.

Expansins are cell wall proteins that are classified as four sub-families, including α-expansin, β-expansin, expansin-like A and expansin-like B. They are required in almost all plant development aspects, from germination to fruiting, such as seed germination, root growth, stem elongation, leaf enlargement and fruit ripening (*Marowa, Ding & Kong, 2016*). These expansin proteins, which induce cell wall extension in plants, were first discovered in cucumber hypocotyls (*McQueen-Mason, Durachko & Cosgrove, 1992*). Expansins are able to loosen and soften plant cell walls, allowing cell expansion because they have the ability to non-enzymatically induce a pH dependent relaxation of the cell wall (*Cosgrove, 2000*; *Marowa, Ding & Kong, 2016*). Expansins are genes in non-GA pathways that have been reported to be involved in height, stem growth and elongation in several plant species, including Arabidopsis such as *AtEXP3* and *AtEXPA10* (*Kuluev et al., 2012*; *Kwon et al., 2008*), rice such as *OsEXP4* and *OsEXPB3* (*Cho & Kende, 1997*; *Choi et al., 2003*; *Lee & Kende, 2001*), in soybean such as *GmEXPB2* (*Guo et al., 2011*), and wheat such as *TaEXPB23* (*Xing et al., 2009*). In 2003, *Choi et al. (2003)* found that *OsEXP4* sense transgenic rice was taller than control rice while *OsEXP4* antisense transgenic rice was shorter than the control rice. They also proposed that expansins are involved in enhancing growth by mediating cell wall loosening. The full-length genomic DNA sequences of *EgExp4* on Chr. 14 (2,586 bp) and *OsEXP4* (GenBank accession: U85246.1, 1,219 bp mRNA) are 82% identical ($E$ value = 1e−83). The full-length genomic DNA sequences of *EgExp4* matched with *Elaeis guineensis* expansin-A2 (LOC105057201, 1,189 bp of mRNA length) 100%. This gene contains three exons, positioned at 1,005–1,119 (Exon1), 1,304–1,616 (Exon 2) and 1,706–2,386 (Exon 3) of the full-length genomic DNA sequence of *EgExp4*. The mEgExp_SNP118 marker is on position 118 of the full-length genomic DNA sequence of *EgExp4* (2,586 bp), on the 5′ UTR of the sequence that is expected to be the *EgExp4* promoter region. After analyzing this 5′ UTR sequence

(1,043 bp) using PlantCARE, a database of plant cis-acting regulatory elements and a portal to tools for *in silico* analysis of promoter sequences (*Lescot et al., 2002*), no motif was located in this SNP position of the sequence. This suggests that this SNP may associate with height either by linkage or functional association.

To date, based on QTL and association studies by using SSR and SNP markers for stem height, height QTLs or height associated genes are positioned in several chromosomes. By using SSR markers, QTLs were located on Chr. 1, 2, 4, and 10 (*Billotte et al., 2010*), and Chr. 16 (*Lee et al., 2015*). By using SNP markers, QTLs were located on Chr. 2 (LG4) and LG 7 (*Yaakub et al., 2020*) Chr. 10, 14, and 15 (*Pootakham et al., 2015*) and Chr. 1, 4, 7, 8, 9, 11, and 15 (*Teh et al., 2020*). Height associated SNP and SSRs targeting genes, including IAA-amido synthetase (*Ong et al., 2018*), growth regulating factor (*EgGRF1*) (*Somyong et al., 2020*) and DELLA (*EgDELLA1*) (*Somyong et al., 2019*) were on Chr. 1, 10 and 14, respectively. In this work, SNP targeting expansin gene (*EgExp4*) (LOC105057201) was on Chr. 14 position 4,625,680–4,627,061 (4.6 Mb). This position was far from the previous height QTL interval between markers EgSNPGBS16523 (position 5.9 Mb) and EgSNPGBS16582 (position 7.5 Mb) on Chr. 14 (*Pootakham et al., 2015*) about 1.3–2.9 Mb.

Because height is a quantitative trait that is controlled by several genes and environment, several SSR and SNP markers in MAS may need to be evaluated in various oil palm populations originating from different ancestry. Some markers might be used across several populations originating from the same ancestry. Based on previous work (*Somyong et al., 2019*, *2020*) and this work, we have identified three different genes, including *EgDELLA1*, *EgGRF1* and *EgExp4*. Therefore, the different approaches help to identify different genes influencing height in the same population. In addition, the correlation of height with oil palm yield was weak to moderately positive in oil palm populations (*Somyong et al., 2019*). This suggests that selecting semi-dwarf palms may contribute to high yield in oil palm and that lowering the height of oil palm will not significantly affect the growth and the production of oil. Consequently, improvement of a dwarf oil palm with yield component traits is challenging for oil palm breeders.

## CONCLUSIONS

PacBio SMRT sequencing was used to identify variations between short oil palm and tall oil palm groups, from the GT population, which include 20 individuals with the shortest phenotypes and 20 individuals with the tallest phenotypes, respectively. To perform amplicon sequencing, height-related genes, including *EgDELLA1*, *EgGRF1*, *EgGA20ox1*, *EgAPG1* and *EgExp4* genes, were amplified by two step-PCR, and the amplified products were used for barcode library preparation and PacBio SMRT sequencing. Twelve variations were identified from *EgDELLA1, EgGRF1, EgGA20ox1 and EgExp4*. In this study, only four SNP variations were confirmed in the GT population (180 oil palm individuals). After, three polymorphic SNP markers (mEgDELLA1_SNP2100, mEgDELLA1_SNP2248 and mEgExp4_SNP118) were found, only mEgExp4_SNP118 was significantly associated with height in the oil palm plants, while the other two polymorphic SNPs were not associated with height in the same population. This finding provided the potential marker in MAS for speeding up the oil palm breeding process.

However, the contribution of this SNP position to height was not determined. We suggest that it may contribute to height either by linkage or functional association. More work on a functional study of *EgExp4* will help to disclose this contribution in oil palm.

## ACKNOWLEDGEMENTS

We also thank the Thailand Research Organizations Network (NRON) managed by ARDA.

### Funding

This research was supported by the Agricultural Research Development Agency (ARDA), Thailand (grant numbers: PRP6105020780 and PRP6305030940). The funders had no role in study design, data collection and analysis, decision to publish, or preparation of the manuscript.

### Grant Disclosures

The following grant information was disclosed by the authors:
Agricultural Research Development Agency (ARDA), Thailand: PRP6105020780 and PRP6305030940.

### Competing Interests

The authors declare that they have no competing interests.

### Author Contributions

- Suthasinee Somyong conceived and designed the experiments, performed the experiments, prepared figures and/or tables, authored or reviewed drafts of the paper, and approved the final draft.
- Phakamas Phetchawang performed the experiments, prepared figures and/or tables, and approved the final draft.
- Abdulloh Kafa Bihi performed the experiments, prepared figures and/or tables, and approved the final draft.
- Chutima Sonthirod analyzed the data, prepared figures and/or tables, and approved the final draft.
- Wasitthee Kongkachana analyzed the data, prepared figures and/or tables, and approved the final draft.
- Duangjai Sangsrakru performed the experiments, prepared figures and/or tables, and approved the final draft.
- Nukoon Jomchai performed the experiments, prepared figures and/or tables, and approved the final draft.
- Wirulda Pootakham conceived and designed the experiments, authored or reviewed drafts of the paper, and approved the final draft.
- Sithichoke Tangphatsornruang conceived and designed the experiments, authored or reviewed drafts of the paper, and approved the final draft.

## Data Availability

The sequences and raw data are available in the Supplemental Files and at NCBI: PRJNA760254.

## Supplemental Information

Supplemental information for this article can be found online at http://dx.doi.org/10.7717/peerj.13046#supplemental-information.

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
