# Peer review of "A SNP variation in an expansin (EgExp4) gene affects height in oil palm"

_PeerJ, doi:10.7717/peerj.13046_

## Round 0.1 · original submission · Major Revisions

Please address the concerns of all reviewers and revise your manuscript accordingly.

Reviewer 1 ·

Basic reporting

The manuscript describes the identification of a SNP of a height gene, namely EgExp4, that recorded significant association with height trait in oil palm. The structure of the manuscript adheres to the journal format. However, the manuscript requires major English language editing.
In the introduction, the authors can also include previous efforts in improving height in oil palm via conventional breeding method. In my knowledge, Elaeis odora does not exist anymore. The citations are relevant but the authors can also cite more recent publications. Figures and tables and their legends are appropriate.

Experimental design

I have specific comments indicated below
1. Are the trees included in categories tall and short the same for HT1, HT2, HT3 and HT4? Palms included in the research were planted at several locations. The height differences observed could be due to palms that are grown at different locations. This should be investigated further. There could be a requirement to normalized the height measured on all palms across different location and time of measurement.
2. Are the differences between the means of the tall and short categories for HT1, HT2, HT3 and HT4 significant?

Validity of the findings

1. In TASSEL analysis, the authors should explain how p value is determined
2. In Table 2, variant change was reported based on some short/tall palms but not all short/tall palms. This lead to a question whether the variant change is truly associated with height. If the change is truly due to height, it should be observed in all short/tall palms.

Reviewer 2 ·

Basic reporting

Somyong et al. reported the discovery of the SNP variation in expansin (EgExp4) gene that may affect height in oil palm. The story that the author presented is complete, and the biological experiment and in-silico studies were sufficiently designed and performed. Although the authors did not point out the functions of the EgExp4 gene and the contributions of the SNPs to the height, they talked about the potential relationship between cell wall protein expansin and plant growth, inspired by their gene study results.

The English writing is clear and the literature references were well-cited with consistent formatting. The figures and articles were well-made, although the resolutions of the main text figures are low with some of the axis labels being not visible.

Nevertheless, I believe the results presented by the authors will inspire researchers to pay attention to the expansin protein and the EgExp4 gene for breeding semi-dwarf oil palms. I therefore recommend it to be published in PeerJ after making necessary revisions and clarifications (please see my comments in the "additional comments" section).

Experimental design

Please see my comments in the "additional comments" section.

Validity of the findings

Please see my comments in the "additional comments" section.

Additional comments

(1) Why is there a high-intensity band in one of the sample loading wells on the last gel of Figure S3?

(2) Will lowering the height of oil palm affects the growth and in turn, affects the production of oil?

(3) In the sentence “the full-lengths of these genes were successfully amplified by the M13-tagged EgDELLA1-P1, EgGRF1-P2, EgGA20ox1-P2, EgAPG1-P1 and EgExp4-P1 primers (Table S2) in the 1st round of PCR with expected sizes of 3015 bp, 2516 bp, 2759 bp, 2917 bp and 2586 bp, respectively”, how did the author know the exact size of each amplified product? If the result of the 1st round of PCR was evaluated by gel electrophoresis, the authors may want to say something like “with bands close to the expected sizes of 3015 bp, 2516 bp, 2759 bp, 2917 bp and 2586 bp”. This also applies to the sentence “The results of amplification by M13-tagged primers and barcode-tagged M13 primers also had expected sizes of 3015 bp, 2516 bp, 2759 bp, 2917 bp and 2586 bp respectively, as shown in Fig. S3”.

(4) Line 94: For the sentence “concentration and quality of DNA were evaluated by agarose gel electrophoresis and a NanoDrop™ 1000 Spectrophotometer”, the authors may want to flip the order of “concentration” and “quality” since gel electrophoresis is not commonly used for DNA concentration determination.

(5) In the abstract, “by two rounds of PCR amplification” may be used instead of “by two PCR round amplification”.

(6) Line 137: “Python” should be used instead of “Phyton”.

(7) Line 325: Please uncapitalize “T” in “Twelve”.

Reviewer 3 ·

Basic reporting

More interpretation of figures and tables should be included.

Experimental design

More detailed validation design, validation results, and validation discussion should be included.

Validity of the findings

1. In Table 3, the results of previous work and this work were listed. And the previous work confirmed that EgDELLA1 and EgGRF1 was height-associated genes in the GT population, which was mentioned in introduction. However, in this study, only four SNP variations were confirmed in the GT population. And only mEgExp4_SNP118 was significantly associated with height in the oil palm plants. Is there any discussion about those?
2. Is there any other reported data to support the results of this study?
3. What is the innovation of this study?

---

## Round 0.2 · Major Revisions

Please address the remaining issues of the reviewers and amend the manuscript accordingly.

Reviewer 1 ·

Basic reporting

After reading the revised manuscript, I think that the manuscript still requires major English language editing.

Elaeis odora does not exist. In Elaeis genus, there are only 2 species, Elaeis guineensis and Elaeis oleifera. The plant previously known as Elaeis odora was re- classified as Barcella odora.

Experimental design

Generally the approach taken to sequence full length candidate genes linked to height using PAC Bio is a good approach and authors have deposited the data in the public databases. However, authors need to explain in a bit more on the parameters used to determine quality of the sequences and steps taken to avoid wrong calls for SNPs or indels. Also a figure showing the match identity of the candidate genes (likely via blast search for example) would be helpful.

Did the authors include the parental palms in sequencing of the candidate genes? This could have indicated the inheritance of the alleles either from the dura or pisifera palms.

Validity of the findings

The authors should justify the p-value applied to determine significance of marker-trait association, as only limited number of markers/individual palms are used in association analysis (4 markers across 40 samples). These obviously limit the power of statistical analysis.

In TASSEL analysis, can the authors present results from MLM model, as this model is available in TASSEL and is more stringent than GLM? The authors can consider presenting the Manhattan plots showing the significant markers in the manuscript which can be easily generate using TASSEL.
Also can the authors explain why only two time points for height measurement- T1 and T4 showed significant association of the allele with height and not T2 and T3? Is there a possibility that the SNPs are linked to other unmeasured traits in the two segregated groups and association with height was incidental?

Additional comments

In the authors previous publications in 2019 and 2020, they found Della and GRF genes. However, in this study the SNPs linked to height was the Exp gene. Three different studies possibly on the same population uncovers three different genes influencing height, and some explanation will help. Although authors explain that height is a quantitative trait, influenced by several genes but a discussion on why different approaches identified three separate genes influencing the same trait in the same population, would be helpful.

Did the authors attempt to link the INDELS to the trait?

Reviewer 2 ·

Basic reporting

The authors have made necessary revisions based on the reviewer's suggestions. The resolution of Figure 1 has improved and the axis labels have been provided. I recommend this manuscript to be published in PeerJ, with a few comments left:

(1) Since the authors have now provided the explanation that "Height correlation with FFB (fresh fruit bunch yield) and BN (Bunch number) was weak to moderately positive in both the GT (r = 0.254-0.499) and KU (r = 0.335-405) oil palm populations. This suggests that semi-dwarf palm may contribute to high yield in oil palm", they may consider including this in the main text to emphasize the importance of selecting semi-dwarf palms, and inform readers that lowering the height of oil palm will not significantly affect the growth and the production of oil.

(2) It might be worth adding the explanation that the authors have provided to the Figure S3 caption, so readers do not get confused with the high-intensity band in the sample loading well on the last gel.

Experimental design

No comment

Validity of the findings

The authors have better addressed the impact and novelty of their work in the Discussion section of the revised manuscript.

Additional comments

No comment

Reviewer 3 ·

Basic reporting

no comment

Experimental design

no comment

Validity of the findings

no comment

---

## Round 0.3 · accepted · Accept

Critiques were adequately addressed and therefore a revised manuscript is acceptable now.